# Flow of cerebrospinal fluid is driven by arterial pulsations and is reduced in hypertension

Humberto Mestre [1,2], Jeffrey Tithof [3], Ting Du[1,4], Wei Song[1], Weiguo Peng[1,5], Amanda M. Sweeney[1], Genaro Olveda[1], John H. Thomas[3], Maiken Nedergaard[1,5] & Douglas H. Kelley [3]

Flow of cerebrospinal fluid (CSF) through perivascular spaces (PVSs) in the brain is important for clearance of metabolic waste. Arterial pulsations are thought to drive flow, but this has never been quantitatively shown. We used particle tracking to quantify CSF flow velocities in PVSs of live mice. CSF flow is pulsatile and driven primarily by the cardiac cycle. The speed of the arterial wall matches that of the CSF, suggesting arterial wall motion is the principal driving mechanism, via a process known as perivascular pumping. Increasing blood pressure leaves the artery diameter unchanged but changes the pulsations of the arterial wall, increasing backflow and thereby reducing net flow in the PVS. Perfusion-fixation alters the normal flow direction and causes a 10-fold reduction in PVS size. We conclude that particle tracking velocimetry enables the study of CSF flow in unprecedented detail and that studying the PVS in vivo avoids fixation artifacts.

[1] Center for Translational Neuromedicine, University of Rochester Medical Center, Rochester, NY 14642, USA. [2] Department of Neuroscience, University of Rochester Medical Center, Rochester, NY 14642, USA. [3] Department of Mechanical Engineering, University of Rochester, Rochester, NY 14627, USA. [4] China Medical University, Shenyang 110122, China. [5] Center for Translational Neuromedicine, Faculty of Health and Medical Sciences, University of Copenhagen, 2200 Copenhagen, Denmark. These authors contributed equally: Humberto Mestre, Jeffrey Tithof. Correspondence and requests for materials should be addressed to D.H.K. (email: d.h.kelley@rochester.edu)

Cerebrospinal fluid (CSF) aids in the removal of metabolic waste from the brain[1]. The exact anatomical pathways and mechanisms underlying how solutes in the interstitial fluid (ISF) are transported towards CSF remain unclear[2]. Historically, it has been thought that solutes exit the brain along a network of perivascular spaces (PVSs) surrounding cerebral arteries, against the direction of blood flow[3,4]. Recent in vivo experiments in rodents have shown the opposite: CSF enters the brain along arterial PVSs, and this flow plays a vital role in driving the clearance of amyloid-β (Aβ) from the ISF at more downstream locations[1]. In both cases, indirect experimental evidence suggests that fluid within PVSs is transported via bulk flow[5–7] and possibly driven by arterial pulsations derived from the cardiac cycle[8–11]. However, some have argued that PVS flow is slow and has little physiologically importance, while others contest that physiologically relevant flow occurs outside the PVS, within the basement membrane of the vessels[2,3,12]. Moreover, several fluid dynamical modeling studies have questioned the feasibility of arterial pulsation-driven flow and have arrived at conflicting results in terms of the magnitude and the overall direction of perivascular CSF flow[13–15]. Magnetic resonance imaging has made recent strides in measuring diffusion coefficients of the PVSs in rats and has shown that the heartbeat dramatically increases the movement of fluid[16]. Nevertheless, this technique lacks sufficient temporal and spatial resolution to determine the velocity fields of CSF in the PVS.

To evaluate fluid motion within the PVS we have adapted in vivo two-photon imaging to allow measurement of CSF flow speeds simultaneously with recordings of cardiac and respiratory cycles. We have also performed synchronized measurements of the artery diameter and heartbeat to determine vessel wall dynamics. The analysis confirms that CSF bulk flow in the PVS is pulsatile, at the same frequency as the cardiac cycle, and in the same direction as blood flow[1,11]. Our results are highly consistent with a fluid transport mechanism—perivascular pumping—wherein vascular wall kinetics directly drive pulsatile CSF bulk flow in the PVS[8]. Furthermore, our in vivo measurements reveal large PVSs surrounding pial arteries, with a cross-sectional area larger than that of the artery, but perfusion fixation collapses these spaces to a tenth of their size in living animals. This finding has significant implications for future experimental and numerical studies and highlights the importance of in vivo measurement. Finally, we show that high blood pressure, a condition that affects close to half of the world's adult population, disrupts the perivascular pump and sharply slows CSF transport in the PVS[17]. Earlier studies have shown that arterial hypertension promotes the accumulation and aggregation of Aβ[18–21]. We speculate that hypertension-induced reduction of PVS fluid transport contributes directly to the associations between arterial hypertension and Alzheimer's disease.

## Results

### Quantifying flow in perivascular spaces. CSF flow was visualized by infusing fluorescent microspheres into the cisterna magna and acquiring images through a sealed cranial window using two-photon microscopy. Microspheres appeared in the PVSs of the cortical branches of the middle cerebral artery (MCA) an average of 292 ± 26 s after the infusion began; these PVSs are known to be primary influx routes for CSF into the brain[22–24]. The infusion did not modify CSF flow speed, and all data sets were collected after the infusion was completed (Supplementary Fig. 1). The motion of the microspheres between consecutive frames, which closely matches the motion of the surrounding fluid, was measured by performing particle tracking velocimetry (PTV; Fig. 1a)[25,26]. In experiments spanning tens of minutes, we

typically tracked more than 20,000 particles (Fig. 1b and Supplementary Table 1). The time-averaged net flow of CSF is in the same direction as the blood flow and varies in space (Fig. 1c). It is fast near the centers of the PVS channels, but slow inside arterial bifurcations and near channel walls (Fig. 1d). Interestingly, bifurcations of leptomeningeal arteries are also common sites of amyloid-β accumulation in Alzheimer's disease[27]; slow flow in the bifurcations might allow that accumulation. Velocity profiles are nearly parabolic (Fig. 1e), as expected for laminar channel flow, with zero velocity at the PVS walls, as required by the no-slip boundary condition. Averaging over both space and time, we find the typical flow speed to be 18.7 μm s$^{-1}$, with variation among animals likely due to small differences in their PVS geometries (95% confidence interval 9.4–28 μm s$^{-1}$; Fig. 1f). Measuring typical flow speeds enabled the first-ever calculations of Reynolds and Péclet numbers in PVSs. The Reynolds number is typically ~0.001 (Fig. 1g), implying that the flow is strictly laminar, with inertial forces much smaller than viscous forces. The high Péclet number (~1000, Fig. 1h) demonstrates that advection dominates diffusive transport in the PVS[15].

### Perivascular space size, shape, and change during fixation. By superimposing all particle tracks from each experiment (Fig. 1b), we can estimate the extent of the PVSs: each is ~40 μm wide, comparable to the adjacent artery. Volumetric images using a dextran CSF tracer confirmed this result (Fig. 2a), and orthogonal projections showed two non-connecting PVSs, one on each side of the artery, with tracer excluded from the outer vessel wall (Fig. 2b, c). The PVS size and shape that we observe agree with in vivo measurements published previously (Fig. 2d)[29,28,30] and do not depend on whether PVSs are imaged through a cranial window or a thinned skull (Supplementary Fig. 2). Identifying the specific structures that bound the observed PVSs is a worthy topic for future study.

We did not observe tracers in the basement membranes of the arteries in our in vivo experiments, though prior studies of fixed tissue have observed them there[1,3,22,24]. Thus, we asked whether tracers enter the vessel wall during tissue processing. Perfusion fixation with 4% paraformaldehyde (PFA) caused abnormal retrograde flow and shrunk the PVS (Fig. 2e–h and Supplementary Movie 1), and also moved tracer from the perivascular space into the vessel wall (Fig. 2i, j). In vivo imaging showed that the cross-sectional area of the PVS is on average about 1.4 times that of the artery, whereas fixation reduced this ratio to 0.14 (Fig. 2k).

### Flow pulsing with heartbeat and driven by artery walls. Both the cardiac and the respiratory cycles have been suggested as CSF flow drivers[7–11,28,29]. To identify the mechanisms driving flow in the PVS, we performed two-photon imaging while recording synchronized measurements of the electrocardiogram (ECG) and the respiratory cycle (Supplementary Movie 2). After performing PTV, we computed the spatial root-mean-square velocity ($v_{rms}$) at every instant of time. The frequency of the $v_{rms}$ peaks matches that of the heartbeat (R wave), but not that of respiration (Fig. 3a). Power spectra and spectrograms are nearly identical for $v_{rms}$ and ECG, demonstrating that the temporal behavior of $v_{rms}$ is very closely linked to the cardiac cycle (Supplementary Fig. 3). Probability density functions of the delay times ($\Delta t$) between an ECG/respiration peak and the next $v_{rms}$ peak (Fig. 3c, inset) indicate that the peak in $v_{rms}$ consistently occurs soon after the heartbeat, whereas a respiration peak has no effect (Fig. 3c). We hypothesize that the delay between the ECG peak and the $v_{rms}$ peak corresponds to the time required for the pulse wave generated by each ventricular contraction to reach the brain. The $v_{rms}$ is modulated strongly over the cardiac cycle (~99%) but weakly

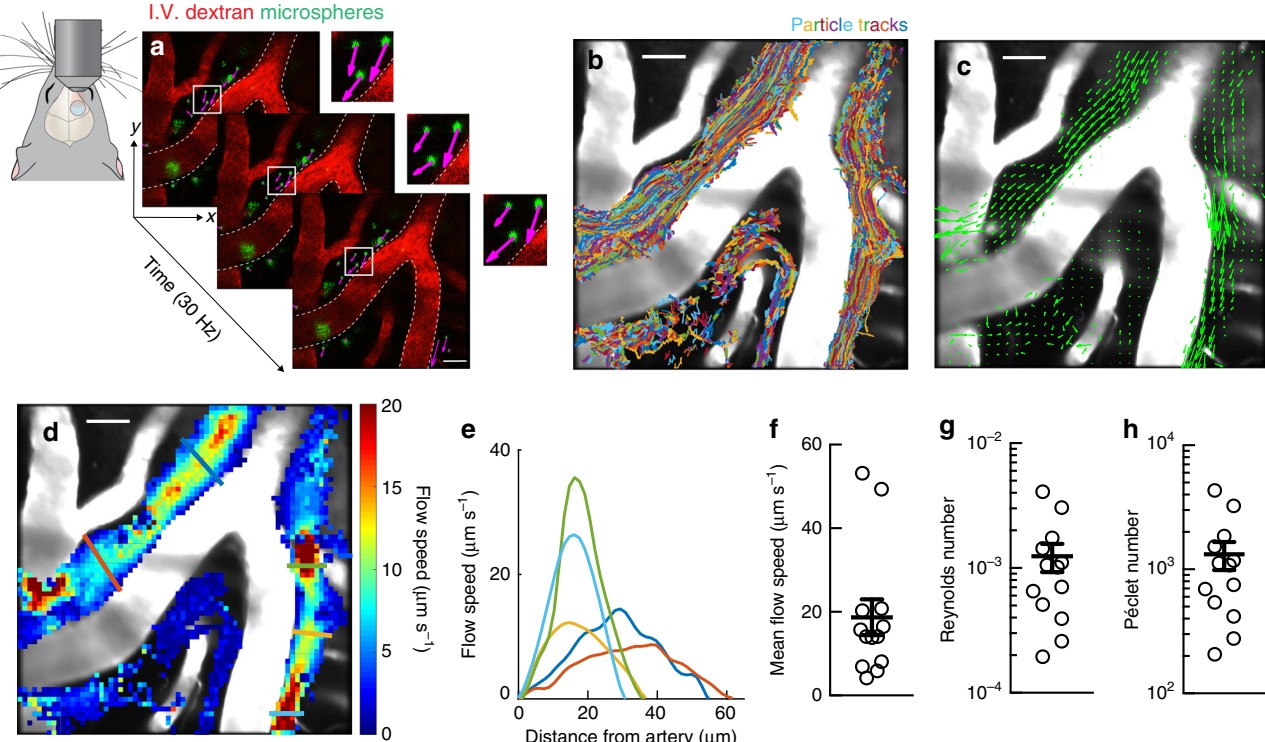

**Fig. 1** CSF in the perivascular space is transported via bulk flow. CSF flow was imaged in live mice through a cranial window using two-photon microscopy. To visualize the CSF, fluorescent microspheres were infused into the cisterna magna. **a** Images for particle tracking velocimetry were acquired at 30 Hz. Blood vessels were labeled with an intravenous (i.v.) dextran, while microspheres appear green; (inset) the magenta arrows show the instantaneous velocity of each microsphere. **b** Superimposed trajectories of tracked microspheres show that particles are transported primarily within large PVSs. Scale bar: 40 μm. **c** The time-averaged velocity field (green arrows) shows that net transport is in the same direction as the blood flow. **d** The local time-averaged flow speed shows that the interior region of the arterial bifurcation is nearly stagnant. Scale bars: 40 μm. **e** Average flow speed profiles plotted as a function of distance from the arterial wall. Colored lines in **d** indicate the location of each profile. **f** Mean flow speeds, **g** Reynolds, and **h** Péclet numbers for the time-averaged flow, mean ± SEM, $n = 13$ mice

over the respiratory cycle (~22%; Fig. 3b, Supplementary Fig. 4a). The modulation depth of the respiratory cycle varies among mice, reaching 45% in some cases (Supplementary Fig. 4b-d). By many measures, the cardiac cycle correlates more closely with CSF flow than the respiratory cycle does and is therefore a more likely driver.

One mechanism by which the cardiac cycle might drive CSF flow is through motion of the arterial wall. We measured the diameter of the pial artery at a fixed location using line scans, while making simultaneous ECG measurements (Fig. 3d), and then computed the normalized average change in artery diameter over the cardiac cycle (Fig. 3e). This average waveform shows a fast diameter increase during the systolic phase and a slow diameter decrease during the diastolic phase. This waveform is substantially different from the sinusoid assumed in most modeling studies[13,30,31]. Computing the derivative of the waveform produces the lateral velocity of the arterial wall (Fig. 3f), which has a maximum of $21.2 \pm 3.7$ μm s$^{-1}$ that occurs $34.5 \pm 7.4$ ms after the ECG peak. The peak in wall velocity and the delay time are consistent with the peak and delay of $\Delta v_{rms}$, suggesting that the local displacement of the arterial wall drives CSF flow via perivascular pumping.

Perivascular pumping might drive more distal flow if the CSF it pumps displaces CSF that is more distal, according to the continuity equation of fluid dynamics. To test this possibility, we tracked particles at more distal locations, including some near arterioles that penetrate into the brain. In those locations, we observed CSF pulsing in synchrony with the heart, with mean flow toward more distal locations (Supplementary Fig. 5), again

suggesting that perivascular pumping is a primary driver. Those observations are consistent with the hypothesis that CSF enters the brain through PVSs around arteries, mixes with interstitial fluid to absorb waste, and leaves the brain through PVSs around veins, presented in prior studies[1,32,33]. Our observation that microspheres infused into the cisterna magna are drawn to PVSs around arteries, and not veins, is also consistent with that hypothesis. The likely reason that flows inside arterial bifurcations are often stagnant is that perivascular pumping generates opposing pressure gradients that sum to approximately zero in these regions. Small differences in perivascular pumping strength between each daughter vessel may drive slow reverse flow (toward more proximal locations) locally in this region. We have observed substantial reverse flow in only one bifurcation region of one experiment; in this case, the daughter branches had significantly different diameters, suggesting the difference in perivascular pumping strength may have been considerable. Flow in the bifurcation, typically slow and complicated, deserves further study; still, flow in PVSs overwhelmingly proceeds toward more distal locations.

**Inducing acute hypertension to alter wall motion and flow.** Having determined that perivascular pumping is a principal driver of CSF flow in the PVS, we next investigated the effect of altering vascular wall dynamics. The amplitude and speed of the vessel wall motion depend on several factors, especially vessel wall stiffness, which depends on local blood pressure: as pressure builds, vascular tone increases, reducing the compliance of the wall[39]. Therefore, hypertension offers a model for altering vessel

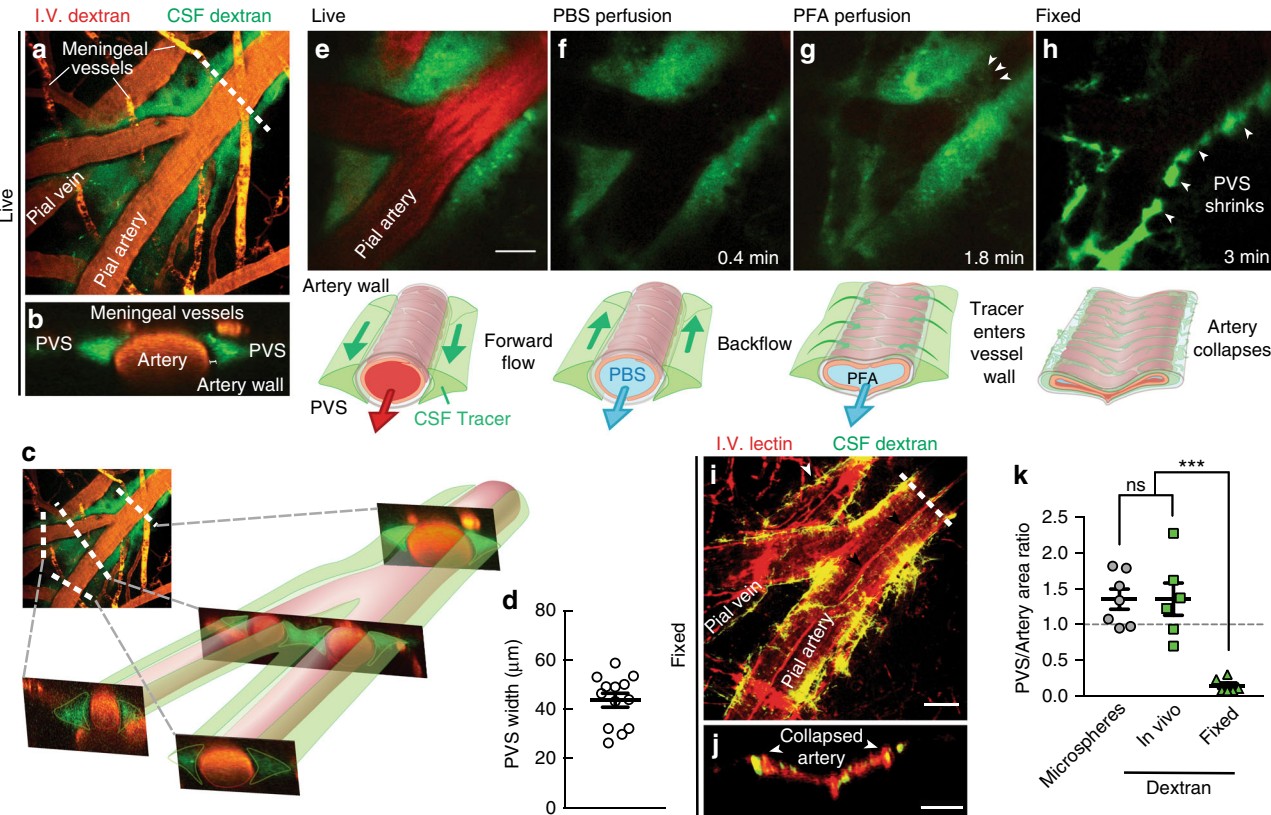

**Fig. 2** Perivascular spaces (PVSs) are larger in vivo and collapse after fixation. **a** Fluorescent dextran in the CSF confirms the size of the PVS in vivo. **b** Cross-section at the dashed line in **a**, showing two PVSs with tracer. **c** Cross-sections proximal, across, and distal to a bifurcation show that the non-connecting PVS structure continues. **d** Average measurements of the width of the PVS from superimposed microsphere trajectories (e.g., Fig. 1b). Mean ± SEM, n = 13 mice. **e** To test whether the PVS and CSF tracer distribution were modified during tissue processing, we perfusion fixed a live anesthetized mouse with **f** phosphate-buffered saline (PBS) followed by **g** 4% paraformaldehyde (PFA) to **h** fix the tissue (Supplementary Movie 1). Diagrams are included below each image indicating our observations. Scale bar: 30 μm. **i** The vasculature was labeled with a lectin in the PBS perfusion solution and the same vessel in the same animal shown in **a** was imaged after fixation. It was necessary to remove dura to image the same vessels in situ due to the shrinkage of the brain. Small arrows point to a fold in the collapsed artery wall and tracer being redistributed around a pial vein. Overlapping lectin and dextran appear yellow. Scale bar: 40 μm. **j** Cross-section at the dashed line in **i** shows that after fixation, the vessel collapses and folds, and the tracer redistributes around the arterial wall. Scale bar: 20 μm. **k** To quantify the size of the PVS compared to that of the artery, we computed the ratios of the areas of the PVS and the adjacent artery for in vivo measurements utilizing tracked microspheres and dextran dye and after fixation (fixed). The lateral area of the PVS is roughly 1.4 ± 0.1 times larger than that of the artery itself in the live mice, and fixation reduces this ratio to about 0.14 ± 0.04. One-way analysis of variance (ANOVA) post hoc Tukey's test, ***P < 0.0002, ns not significant, mean ± SEM, n = 6–7/group

wall dynamics. To induce arterial hypertension, we screened several hypertensive agents and selected an intravenous infusion of angiotensin-II (Ang-II) due to its tight temporal control on increasing mean arterial blood pressure (Fig. 4a, b). Infusion of Ang-II did not change the average arterial diameter (Fig. 4c) or disrupt the blood–brain barrier (BBB; Supplementary Fig. 6). We recorded arterial line scans at proximal and distal locations (Fig. 4d) and measured the average vessel wall motion and velocity. Both the arterial waveform and the wall velocity changed substantially at distal locations during hypertension (Fig. 4e–g). These changes are most likely caused by the arteries becoming stiffer, such that waves are damped less; this primarily affects smaller, distal arteries that have thinner, less muscular walls and are therefore too weak to maintain flexibility while supporting increased blood pressure.

To test whether these changes in the vessel wall dynamics affect the efficiency of the perivascular pump, we imaged CSF flow and simultaneously recorded ECG, respiration, and arterial blood pressure before, during, and after administering Ang-II (Supplementary Movie 3). The average flow speed (Fig. 5a) was reduced by ~40% during Ang-II-induced hypertension (Fig. 5b). This decrease in flow speed was independent of the time after infusion

(Supplementary Fig. 7a-b). Also, microsphere trajectories show more backflow during each cardiac period in acute hypertension (Fig. 5a, insets). To quantify changes in the instantaneous flow rate, we computed the downstream component of every velocity measurement (Fig. 5c, d). Hypertension increases backflow (negative downstream velocity) by 21% relative to controls (Fig. 5e and Supplementary Fig. 7c). A small percentage of the microspheres aggregate and adhere to the PVS wall, remaining in the imaging field for the duration of the experiment; however, hypertension did not increase the number of these stuck particles compared to the controls (Supplementary Fig. 8).

## Discussion

Our measurements provide the first detailed evidence of perivascular pumping in PVSs of the rodent brain (Supplementary Movie 4). Although we cannot exclude the possibility that additional mechanisms may drive CSF through PVSs, we note that any steady external pressure gradient (resulting from production of CSF, for example) must typically be weaker than perivascular pumping, since the resulting flow often reverses direction during each cardiac cycle.

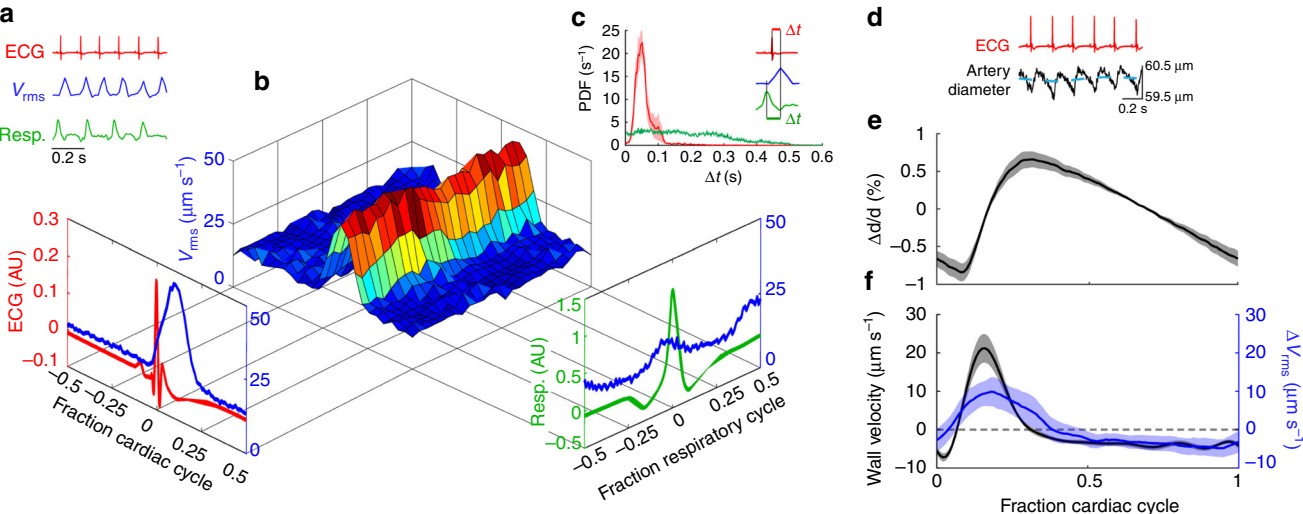

**Fig. 3** CSF in the perivascular space is pumped by arterial wall motions. **a** Representative traces of ECG (red curve), root-mean-square velocity $v_{rms}$ (blue curve), and respiration (green curve). **b** The $v_{rms}$ conditionally averaged over the cardiac and respiratory cycles[38]. The protruding plots show $v_{rms}$ averaged over the cardiac (left) or respiratory (right) cycle alone. **c** (inset) An example illustrating how $\Delta t$, the delay time between adjacent peaks in the $v_{rms}$ and ECG/respiration, is calculated. Average probability density functions of $\Delta t$ for ECG (red) and respiration (green), with shaded regions indicating standard error of the mean (SEM). While there is no clear trend for the occurrence of a peak in $v_{rms}$ after a respiration peak, the $v_{rms}$ peaks are most likely to occur soon after an ECG peak. **d** Synchronized measurements of the ECG (red curve) and artery diameter (black curve), obtained from transversal line scans of the pial arteries; the blue dashed line is a rolling average of the artery diameter. **e** The normalized average change in the artery diameter (i.e., the shape of the arterial wall traveling wave) averaged over the cardiac cycle. Mean ± SEM, $n = 7$ mice. **f** The arterial wall velocity (black curve) obtained by calculating the average derivative of the change in artery diameter over the cardiac cycle, and $\Delta v_{rms}$ (blue curve) obtained by calculating the difference in $v_{rms}$ from its mean over the cardiac cycle. Mean ± SEM, $n = 7$ mice. The maximum arterial wall velocity ($21.2 \pm 3.7\ \mu m\ s^{-1}$, $n = 7$) agrees well with the peak $\Delta v_{rms}$ value ($9.9 \pm 3.8\ \mu m\ s^{-1}$, $n = 8$) and occurs $34.5 \pm 7.4$ ms after the ECG peak, indicating that perivascular pumping is most likely the principal driving mechanism

Our observations of PVSs collapsing during perfusion fixation may explain prior discrepancies in PVS size estimates, and have implications for future experimental studies. During collapse, the PVS cross-sectional area decreases by a factor of 10; CSF flows vigorously toward the cisterna magna, in the opposite direction observed in live animals; and fluorescent tracers move from the PVS to the basal lamina between the vascular smooth muscle cells. Thus, future experimental studies should be mindful of the facts that the PVS changes dramatically during fixation and that the distribution of CSF tracers in histological sections does not reflect their locations when the animals were alive. The PVS sizes we observe have implications for future modeling studies, since width is a sensitive parameter in fluid dynamical models; hydraulic flow resistance scales with the inverse fourth power of the width. The exact path by which CSF flows into the brain is an important topic of future research; its study will require new methods because particles large enough to be tracked individually are not transported along the PVSs of penetrating arterioles, as demonstrated by our experiments and prior studies[11].

We find that the perivascular pump becomes less efficient in acute arterial hypertension due to increased backflow, likely arising from changes in the vessel wall dynamics. In particular, acute arterial hypertension leads to a change in the waveform of the arterial wall motion and a greater negative wall velocity, which may explain the increase in backflow. Furthermore, these blood pressure-dependent changes in waveform and wall velocity are larger at more distal locations, which is a common feature of arterial hypertension[34]. Since CSF flow drives waste clearance in the brain, and prior studies in both rodents and humans have demonstrated that hypertension is correlated with amyloid-β accumulation[18–21], our study offers evidence for a novel causal mechanism: arterial hypertension induces a change in vessel dynamics that reduces perivascular pumping, decreasing the net flow of CSF in PVSs. Conservation of mass dictates that such a

decrease in the PVS flow will lead to a similar decrease throughout the glymphatic system, a brain clearance pathway shown to play a role in amyloid-β removal[1]. Hence hypertension, a known risk factor for Alzheimer's disease, may reduce parenchymal waste transport. The decrease in CSF flow that we observe in acute arterial hypertension could be explained by stiffening of artery walls, which suggests that the same mechanism may be relevant for chronic hypertension and arteriosclerosis. Future studies, using methods that better model human essential hypertension (spontaneous hypertensive mice or slow pressor angiotensin-II infusion), will provide insight into how vascular remodeling in response to long-lasting elevation of blood pressure affects PVS fluid transport.

## Methods

**Animals and surgical preparation**. All experiments were approved by the University Committee on Animal Resources of the University of Rochester Medical Center (Protocol No. 2011-023) and an effort was made to minimize the number of animals used. Male C57BL/6 mice, 8–12 weeks of age (Charles River), were anesthetized with ketamine/xylazine (100/10 mg kg⁻¹, intraperitoneally). Body temperature was maintained at 37.5 °C with a rectal probe-controlled heated platform (Harvard Apparatus). A cranial window was prepared over the MCA vascular territory on the right anterolateral parietal bone leaving dura mater intact. The cranial window was sealed with agarose (0.8% at 37 °C) and a glass coverslip to prevent intracranial depressurization[35]. For a subset of experiments, instead of a cranial window, mice received a thin-skull preparation[9,36]. Afterwards, a cannula was placed in the cisterna magna and red fluorescent polystyrene microspheres (FluoSpheres™ 1.0 μm, 580/605 nm, 0.25% solids in artificial CSF (aCSF), Invitrogen) were briefly sonicated and infused at 2 μl min⁻¹ for 5 min[1].

**Measurement of vital signs**. ECG and respiratory rate were acquired at 1 kHz and 250 Hz, respectively, using a small animal physiological monitoring device (Harvard Apparatus). Arterial blood pressure was measured through an arterial catheter placed in the femoral artery and connected to a pressure transducer and monitor (World Precision Instruments). The signals were digitized and recorded with a DigiData 1550A digitizer and AxoScope software (Axon Instruments).

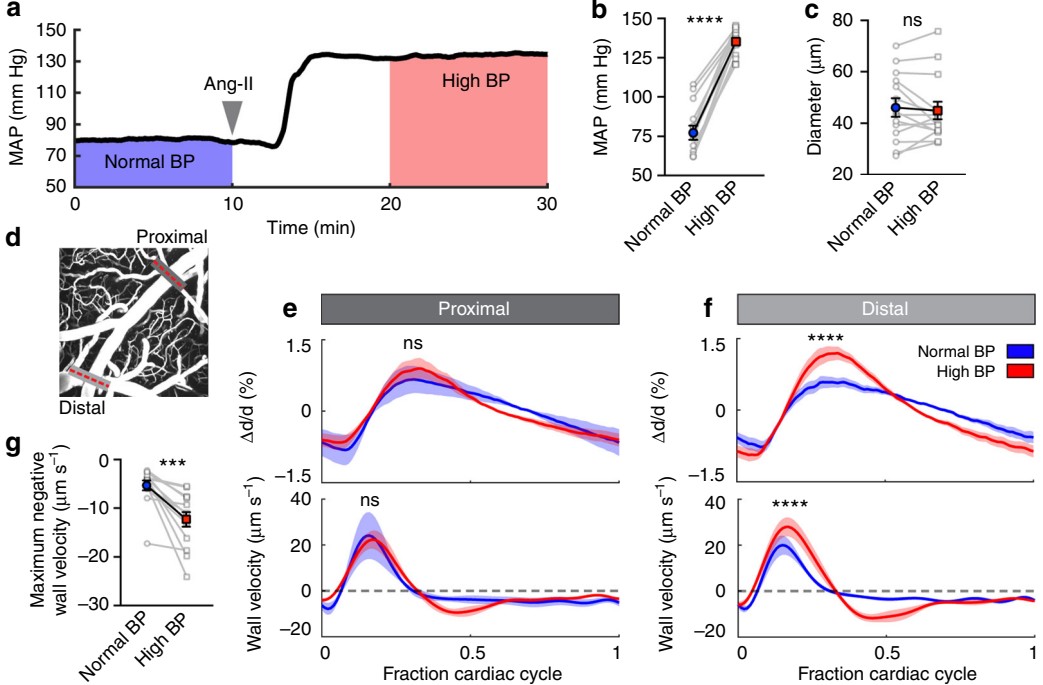

**Fig. 4** Acute arterial hypertension changes vessel wall dynamics. **a** We induced acute arterial hypertension with an intravenous infusion of angiotensin-II (Ang-II). **b** Average blood pressure over the two intervals indicated in (**a**). Ang-II increased mean arterial blood pressure (MAP) by 75% (77.1 ± 4.5 to 134.9 ± 2.2 mm Hg). Paired $t$-test, ****$P < 0.0001$, mean ± SEM, $n = 15$ mice. **c** Average artery diameter before and after Ang-II infusion. Paired $t$-test, $P = 0.5500$, ns not significant, mean ± SEM, $n = 14$ (2 arteries/7 mice). **d** Line scans to quantify artery diameter were acquired at a proximal location of the pial artery and at a distal location before and after Ang-II infusion (gray lines). **e**, **f** The average change in artery diameter (top panel) and the average arterial wall velocity (bottom panel) in normal and high blood pressure, measured at the **e** proximal (dark gray) and **f** distal (light gray) locations indicated in **d**. Inducing high blood pressure alters the form of the arterial traveling wave and makes arteries expand and contract faster; the effects are more prominent at more distal locations. Two-way repeated measures ANOVA. ***$P < 0.0001$, ns not significant, mean ± SEM. **g** High blood pressure increases maximum negative wall velocity. Paired $t$-test, $P = 0.0003$, mean ± SEM, $n = 14$ (2 arteries/7 mice)

**Acute arterial hypertension induction**. Angiotensin-II (Tocris) was dissolved in NaCl 0.9% and infused into a femoral vein catheter at 5 ng g$^{-1}$ min$^{-1}$ at a volumetric rate of 1 µl min$^{-1}$ using a syringe pump (Harvard Apparatus) for the duration of the experiment. Mice with less than a 20 mm Hg increase in blood pressure were excluded from the analysis. Control mice received an intravenous infusion of NaCl 0.9% at 1 µl min$^{-1}$.

**In vivo two-photon laser scanning microscopy**. Two-photon imaging was performed using a resonant scanner B scope (Thorlabs) with a Mai Tai DeepSee HP Ti:Sapphire laser (Spectra Physics) or a Chameleon Ultra II laser (Coherent) attached to a galvo confocal scanning system (Fluoview 300, Olympus). A water-immersion 20× objective (1.0 NA, Olympus) was used on both systems. Intravascular fluorescein isothiocyanate–dextran (FITC–dextran, 2,000 kDa) and red microspheres were excited at an 820 nm wavelength. Images were acquired at 30 Hz (ThorSync software) simultaneously with physiological recordings (3 kHz, ThorSync software). Prior to infusion of the microspheres, a volumetric image of the vascular topology was acquired. Segments of the MCA were identified as large caliber surface vessels (50–60 µm diameter) that started at the anterolateral aspect of the window and traveled towards the midline. Direction of blood flow was confirmed before and after imaging using line scans.

**Correlative imaging**. To confirm the size of the PVS, a separate group of anesthetized mice received an intracisternal injection of Texas-Red dextran (2000 kDa, 0.5% in aCSF, Invitrogen) instead of microspheres. The dextran tracer was delivered using the same infusion paradigm as described above. After tracer reached the PVS of the cortical arteries, volumetric images were acquired for cross-sectional reconstruction. Mice were then perfused with a phosphate-buffered saline (PBS) solution with an Alexa Fluor 488 wheat germ agglutinin lectin (15 µg ml$^{-1}$, Invitrogen) through an arterial catheter. Afterwards, mice were perfusion-fixed with 4% PFA. After fixation, the coverslip and the dura mater were removed due to tissue shrinking and images were acquired as described above.

**BBB permeability assay**. FITC-conjugated dextran (1%; 70 kDa; Sigma-Aldrich) in normal saline (4 ml kg$^{-1}$) was injected via a femoral vein catheter. The dextran was allowed to circulate for 30 min while mice received either an Ang-II or NaCl

0.9% infusion (1 µl min$^{-1}$). The brains were harvested, sectioned, and FITC extravasation was imaged and quantified (see below).

**Tissue processing and imaging**. For assessment of BBB permeability, mice were transcardially perfused with ice-cold 0.1 M PBS (pH 7.4, Sigma-Aldrich) followed by 4% PFA. Brain tissue was carefully dissected away from the skull and dura then post-fixed overnight in 4% PFA at 4 °C. For correlative imaging, mice were perfusion fixed as before but Alexa 488-conjugated lectin from wheat germ agglutinin (15 µg ml$^{-1}$; Invitrogen) was added into the ice-cold PBS solution prior to the PFA perfusion step.

**Ex vivo fluorescence imaging**. Coronal slices (100 µm thickness) were obtained using a calibrated vibratome (VT1200S, Leica). Beginning at the anterior aspect of the corpus callosum, one section was collected every 5 sections until a total of 6 sections had been acquired for each animal. Brain sections were mounted with ProLong Gold Antifade with 4′,6-diamidino-2-phenylindole (DAPI; Invitrogen). Coronal sections were imaged at ×4 magnification using a montage epifluorescence microscope (BX51 Olympus and CellSens Software). Exposure times and magnifications were kept the same for all groups.

**Image processing**. The images obtained from two-photon microscopy are 16-bit with two channels (red and green), each with spatial dimensions of 512 × 512. The green channel captures the FITC–dextran in the vasculature, while the red channel captures the fluorescent microspheres flowing in the perivascular spaces. Note that in the figures presented in this study, we have interchanged the two channels to improve clarity (red vasculature is more intuitive), but the discussion here will refer to the two channels by their accurate, pre-interchanged colors. The first step of image processing is to perform image registration. A time series of rigid translations (no rotation or deformation) is calculated to an accuracy of 0.2 pixel using an efficient algorithm in MATLAB[37]. The translations are inspected for outliers (where the algorithm has failed due to an erroneous correlation) which are manually corrected via linear interpolation. Images from both the red and green channels are then sequentially read and rewritten with the rigid translations applied; the edges of each image are padded with zero-valued pixels such that all final images in each time series have the same spatial dimensions. One mouse was excluded because image registration could not be performed effectively. The second

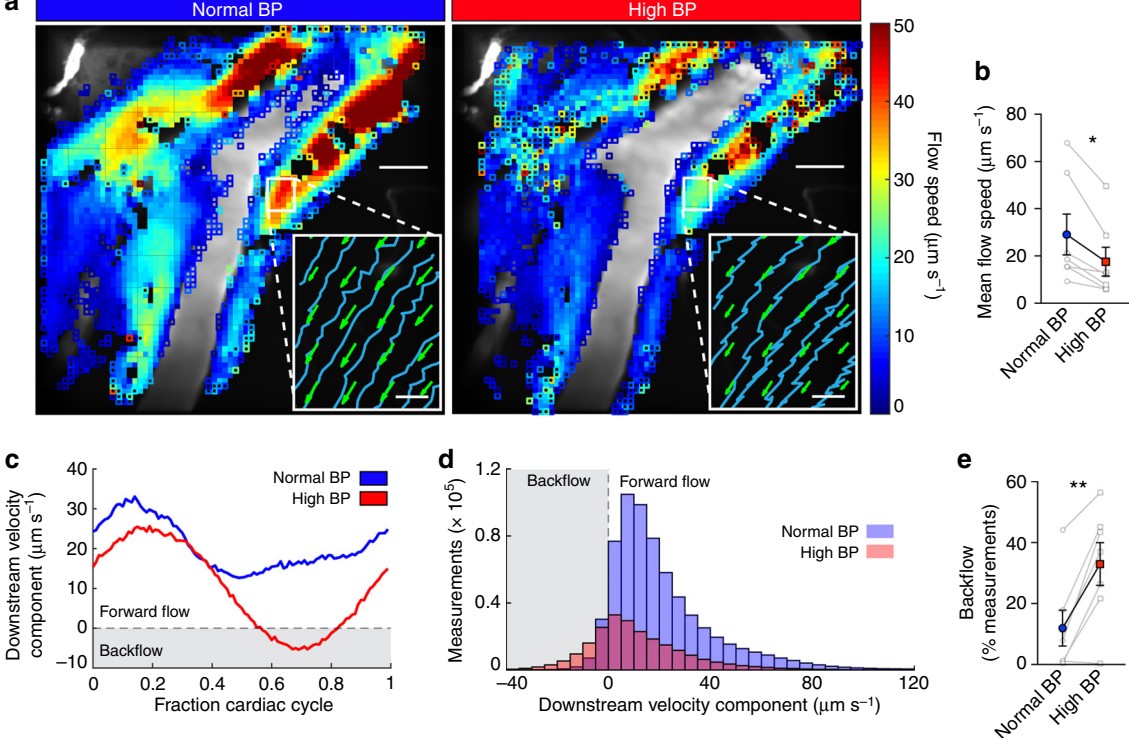

**Fig. 5** Acute arterial hypertension reduces net CSF flow in the perivascular space by increasing backflow. **a** Local time-averaged flow speeds for the normal and high blood pressure time intervals. Closed squares indicate regions with at least 20 measurements, open squares fewer than 20. The flow speed is substantially reduced in hypertension. Scale bars: 40 μm. (Insets) Sample microsphere trajectories (blue curves) and mean flow velocities (green arrows) in the region indicated by the white box. Microsphere trajectories show increased backflow in hypertension. Scale bars: 5 μm. **b** Measurements of the mean flow speed for normal and high blood pressure. Paired *t*-test, *$P = 0.0119$, mean ± SEM, $n = 7$. **c** The average downstream velocity component plotted as a function of the fraction of the cardiac cycle, averaged over all velocity measurements obtained in a small region, for normal and high blood pressure (measurements correspond to the mouse shown in **a**). Hypertension is characterized by backflow over a short segment of the cardiac cycle. **d** Histograms of the downstream velocity component computed for intervals of normal and high blood pressure for the mouse shown in **a**. We obtain fewer high blood pressure measurements ($2.3 \times 10^5$, compared to $6.6 \times 10^5$ for normal blood pressure) because slower flow brings fewer particles into the field of view. The distribution shifts left indicating an increase in backflow when blood pressure is high. **e** The percent of total measurements which correspond to backflow (i.e., negative measurements of the downstream velocity component) calculated for normal and high blood pressure. Paired *t*-test, **$P = 0.0035$, mean ± SEM, $n = 7$. In hypertension, slower net flow and increased backflow are consistent with an alteration of the perivascular pump, characterized by the changes to the arterial waveform and wall velocity

and final step of image processing was to mask regions with stagnant microspheres in the red channel. This was achieved by computing the average of the entire time series of registered red channel images, then applying a threshold to generate a binary mask. The binary mask was used as input to the particle tracking velocimetry software (discussed below) such that regions above the threshold were completely masked from any velocity measurements. This masking technique ensured that stagnant particles (typically stuck to the boundaries of the perivascular space) did not erroneously decrease the measurements of the flow speed. BBB permeability to FITC–dextran was quantified from ex vivo coronal sections by a blinded investigator. Each slice was analyzed for mean pixel intensity and the average was computed for all 6 sections taken from one brain. Full width at half maximum was quantified by measuring the intensity distribution across the center of 20 randomly selected particles from both types of cranial windows. All images were analyzed using ImageJ software (National Institutes of Health, imagej.nih.gov/ij/).

**Particle tracking velocimetry.** To obtain measurements of the speed of cerebrospinal fluid flow, we have quantified the motion of the microspheres in each registered time series of images using an automated PTV software in MATLAB[25,26]. This algorithm individually locates each particle with subpixel accuracy, tracks its location throughout the time series of images, and calculates its velocity in each frame. Hence, for each particle, a time series of positions and velocities is obtained. To further ensure that no stagnant particles were included in our measurements, we computed the average speed along each particle track by dividing the total displacement by the total number of frames in which that particle was tracked; particle tracks with average speeds lower than a given threshold (typically about 2 μm s$^{-1}$) were excluded from the analysis. Supplementary Table 1 shows the total number of particle tracks and the total number of velocity measurements obtained for each of the 13 mice for which PTV has been performed. To

compute the time-averaged flow velocity and speed, the domain was divided into approximately 70 × 70 boxes of resolution 7.5 × 7.5 pixels each. All individual velocity measurements for a given time interval were then binned and averaged according to their box positions. Average flow speeds were computed using only bins which had sufficient measurements (at least 20). Stagnant or nearly stagnant regions were excluded when calculating the changes in flow speed. Mice with fewer than 80 non-stagnant bins with sufficient measurements were excluded from the analysis. To calculate $v_{rms}$, we computed the spatial root-mean-square of all speed measurements obtained at each instant of time.

**Calculation of Reynolds and Péclet numbers.** The Reynolds number is calculated as:

$$\mathrm{Re} = \frac{UL}{\nu},\tag{1}$$

where $U$ is the spatial mean of the magnitude of the time-averaged flow velocity (e.g., the mean of measurements shown in Fig. 1d), $L = 4.4 \times 10^{-5}$ m is the average PVS width that has been measured (Fig. 2d), and $\nu = 0.697 \times 10^{-6}$ m$^2$ s$^{-1}$ is the kinematic viscosity of water at 36.8 °C (a good approximation for cerebrospinal fluid). The Péclet number is calculated as:

$$\mathrm{Pe} = \frac{UL}{D},\tag{2}$$

where $D = 6.55 \times 10^{-13}$ m$^2$ s$^{-1}$ is the diffusion coefficient for 1 μm spherical particles in a liquid at low Reynolds number. The diffusion coefficient is calculated

using the Stokes–Einstein equation:

$$D = \frac{kT}{6\pi\eta r}, \qquad (3)$$

where $k = 1.38 \times 10^{-23}$ m$^2$ kg s$^{-2}$ K$^{-1}$, $T = 310$ K is the body temperature of a mouse, $\eta = 6.93 \times 10^{-4}$ kg m$^{-1}$ s$^{-1}$ is the dynamic viscosity of water at 36.8 °C, and $r = 0.5 \times 10^{-6}$ m is the microsphere radius.

**Artery diameter and wall velocity measurements.** All line scans were recorded transverse to the MCA. The line scans were recorded at about 600 Hz with synchronized ECG measurements, from which we obtained about 120 artery diameter measurements per cardiac cycle (depending on the heart rate). Line scans at each location were recorded for at least 45 s. For each line scan time series, the outer edges of the artery were located with subpixel accuracy at each instant of time by performing a cubic interpolation of the pixel intensity profile onto a finer grid and identifying the two locations where the pixel intensity crossed a given threshold. For each time series, the threshold value was chosen to approximately coincide with the region of the steepest gradient in the pixel intensity profile. The artery diameter at each instant of time was calculated by measuring the distance between these two locations. Using the synchronized ECG measurements, the cardiac cycle was defined by sequential peaks of the R wave. The change in artery diameter was then calculated as the difference between the instantaneous artery diameter and a two-cardiac-cycle moving average of the artery diameter. The changes in artery diameter measurements were then binned and averaged according to the fraction of the cardiac cycle to obtain the mean change in the artery diameter $\Delta d$; this quantity was then normalized by dividing by the average artery diameter $d$. To measure the artery wall velocity, the mean change in artery diameter measurements (in dimensional distance versus time units) for each animal was fit with a high-order Fourier series in MATLAB and the derivative was calculated analytically.

**Downstream velocity component.** The downstream velocity component is calculated as $\mathbf{u} \cdot \widehat{\mathbf{u}}_{\mathrm{avg}}$, where $\mathbf{u}$ is the instantaneous particle velocity and $\widehat{\mathbf{u}}_{\mathrm{avg}}$ is a field of unit vectors computed from the time-averaged velocity field (see, e.g., Fig. 1c).

**PVS/artery area ratio.** Each area used in computing an area ratio was obtained in MATLAB by generating a binary image of the relevant space (perivascular space or artery) and summing the binary image. To generate a binary image for a region which was visualized via fluorescent dye, a carefully chosen threshold was applied to each given image. To generate a binary image indicating the area of the perivascular spaces using microsphere trajectories, we followed the following procedure: all microsphere locations over the entire time series were superimposed onto one large set of coordinates. These (subpixel) coordinates were then rounded to the nearest pixel, and each microsphere was inflated to encompass a $3 \times 3$ pixel region. This set of coordinates was then used to construct the binary image. Before the area ratio was computed, a region of interest was manually chosen for the calculation such that we excluded the vasculature and perivascular spaces not associated with the middle cerebral artery. Furthermore, any region of overlap between the two binary images was subtracted from the perivascular binary image so that the area ratio would characterize the lateral extent of the perivascular space beyond the artery. Three-dimensional surface reconstructions and orthogonal views of the PVS were done using Imaris (Bitplane).

**Statistical analysis.** All statistical analyses were performed on GraphPad Prism 7 (GraphPad Software). No sample size calculations were done due to the lack of an existing effect size estimate. An effort was made to have $n > 5$ for each experiment. Statistical tests were selected after evaluating normality (D'Agostino–Pearson omnibus test) or after visualization of the distribution of data when the sample size did not allow for testing. In the latter cases, both parametric and nonparametric tests were performed and, in all cases, yielded the same result. All testing was two-tailed and exact $P$ values were calculated at a 0.05 level of significance and stated in the figure legends. Animals were not randomized and blinding was done where possible, but was generally not practical for PTV given that the investigator required access to the arterial blood pressure data for analysis.

**Code availability.** All relevant codes are available from the authors.

## Data availability

All relevant data are available from the authors. A reporting summary for this article is available as a Supplementary Information file.

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

## Acknowledgements

This work was supported by the National Institute of Neurological Disorders and Stroke (R01 NS100366 to M.N.), the National Institute of Aging (RF1 AG057575-01 to M.N., J.H.T., and D.H.K.), Foundation Leducq (FLQ 12CVD01 to M.N.), and the European Union's Horizon 2020 Research and Innovation Programme (SVDs@target) (666881 to M.N.). We would like to thank Dan Xue for assistance with illustrations.

## Author contributions

H.M., M.N., J.T., D.H.K., and J.H.T. conceived and guided the project; H.M. and M.N. designed the experiments; H.M., T.D., W.S., W.P., A.M.S., and G.O. performed the experiments; J.T. and D.H.K. performed particle tracking velocimetry; H.M., J.T., and D.H.K. performed the statistical analysis; and J.T., H.M., J.H.T., D.H.K., and M.N. wrote and edited the manuscript.

## Additional information

**Competing interests:** The authors declare no competing interests.

