## [Peer Review File · Nature Communications]

Reviewers' comments:

Reviewer #1 (Remarks to the Author):

Mestre et al. present a precise study on CSF flow in anesthetized mice. Using particle tracking velocimetry, they show that CSF flow in the perivascular space is driven by the cardiac pressure wave which is transmitted at each cycle to large brain arteries. Raising blood pressure changes the vessel wall dynamics and reduces the net forward CSF flow. The approach is technically elegant, demonstrates clearly that vascular pumping drives CSF in the perivascular space of large surface vessels and support the idea that CSF transport is a new marker that should be monitored in brain pathology. The paper is solid, clearly written, and I have few issues that can be easily addressed:

Major comments

-Fig 1d shows two non-connecting PVSs. This anatomical shape is very intriguing. It is important that the authors verify that a similar shape is observed in the thinned skull preparation, as it could be artefactual and result from the pressure generated by the glass window over the brain.

-The authors limited their study to large surface arteries. I understand that due to technical limitations, particle tracking cannot be easily performed in descending arterioles. However, the authors should extend their observations to small horizontal arteries, right before they dive in brain parenchyma. It will allow to test whether CSF pulsations are rapidly damped or transmitted downstream to the glymphatic system.

Minor comments

-Could the authors indicate the time particles take to reach the MCA PVS?

-In extended Fig 5, could the authors lower the threshold to a value where noise can be observed in control condition? It is necessary to assess the effect of Ang-II.

Reviewer #2 (Remarks to the Author):

CSF movement through the brain is an important but poorly understood process. It is thought to occur through perivascular spaces (PVSs) and is potentially necessary for clearance of metabolic by-products such as amyloid- β . The authors use intravital microscopy of cranial windows in mice to follow the movement of particles injected into the cisterna magna. The authors conclude that the motion proceeds in the same direction as blood flow, and is produced by arterial pulsations within large PVSs at the brain surface. The idea that arterial pumping drives flow is not novel, but direct visualization of the process would be valuable for understanding the process. Unfortunately, the study appears very preliminary, and the PVSs shown in the figures and movies require more characterization.

Specific Comments:

In Figure 1C, D, the 3D shape of the PVS is unexpected; what does the cross section look like farther downstream, just after the bifurcation? It appears that the CFS tracers are not confined within anatomical, annular structures surrounding the arteries, but are instead trapped between the brain

tissue and glass coverslip. Can the authors show that there is a biological structure containing the dextran by IHC or EM?

The direction of the motion would be expected to take the particles away from the heart, into the brain. But the particles supposedly are being drained from the brain. Can the authors explain this discrepancy?

The rationale and interpretation of Fig 1e, f are not clear. Is the yellow color dextran that has been fixed in place? Why is it only at the vessel wall, and why does it appear along the vein (when it is not visible in Fig 1c)?

With a bolus injection of microspheres, how long does it take to clear from the PVS? Are the pharmacokinetics reproducible?

The proposed PVS space seems to be rather large. How do the authors' measurements compare with other literature values for the width of the perivascular space?

How long does it take for the particles to travel from the cisterna magna to the pial surface? Does this transport occur along arterials? Would such transport be anatomically consistent with the blood vasculature of the brain?

Can the authors explain how peristaltic motion would be the driving force for PVS flow, when it cannot extend into the capillaries and venules? How would the particles continue moving down the vascular tree, if the flow relies on peristaltic arterial motion? Some speculation or explanation is needed here.

Responses to Reviewer #1

Mestre et al. present a precise study on CSF flow in anesthetized mice. Using particle tracking velocimetry, they show that CSF flow in the perivascular space is driven by the cardiac pressure wave which is transmitted at each cycle to large brain arteries. Raising blood pressure changes the vessel wall dynamics and reduces the net forward CSF flow. The approach is technically elegant, demonstrates clearly that vascular pumping drives CSF in the perivascular space of large surface vessels and support the idea that CSF transport is a new marker that should be monitored in brain pathology. The paper is solid, clearly written, and I have few issues that can be easily addressed:

We thank Reviewer #1 for these positive comments.

Major comments

-Fig 1d shows two non-connecting PVSs. This anatomical shape is very intriguing. It is important that the authors verify that a similar shape is observed in the thinned skull preparation, as it could be artefactual and result from the pressure generated by the glass window over the brain.

We agree that the shape and size of the perivascular spaces we observe are worth noting, particularly because they are substantially larger than what can be seen *ex vivo* and has often been described in the past. The manuscript now dedicates an entire figure to PVS shape (Fig. 2), including cross-sections both proximal and distal to the MCA bifurcation. The figure also shows the PVS collapsing during fixation. Additionally, we have added Supplementary Fig. 2, which shows PVS shape and CSF flow speed observed using a thinned-skull preparation. The shape and speed are consistent with measurements using cranial windows. The advantage of using cranial windows is improved optical access, as quantified by Supplementary Fig. 2i-k, which shows that radial point spread function is greater with thinned-skull preparations, a possible source of error, as noted previously [Bedussi et al. *J Cerebr Blood F Met* 2017]. Moreover, we include a dilute agarose solution between coverslip and dura to preserve curvature of the meningeal surface during experiments with cranial windows. Finally, our observations of the shape and size of the PVS are consistent with several prior studies [Bedussi et al. *J Cerebr Blood F Met* 2017; Schain et al. *J. Neurosci.* 2017; Coles et al. *Methods*, 2017].

-The authors limited their study to large surface arteries. I understand that due to technical limitations, particle tracking cannot be easily performed in descending arterioles. However, the authors should extend their observations to small horizontal arteries, right before they dive in brain parenchyma. It will allow to test whether CSF pulsations are rapidly damped or transmitted downstream to the lymphatic system.

We thank the reviewer for raising a key point: the fluid pumped through perivascular spaces must go somewhere, and discovering where it goes is an essential challenge for researchers in the field. Considering this point, we performed additional experiments to measure flow speed in perivascular spaces surrounding distal, small pial arteries, including regions adjacent to penetrating arterioles. The results are now included as Supplementary Fig. 5, which shows that

pulsations and similar flow speeds extend along smaller pial arteries, past bifurcations, and right up to penetrating arterioles. Our group is working on identifying novel fluorescent particles that will be able to access the PVSs of penetrating arteries.

Minor comments

-Could the authors indicate the time particles take to reach the MCA PVS?

The manuscript now states that particles reach the MCA PVS 292 ± 26 s after the infusion begins (n = 7 mice).

-In extended Fig 5, could the authors lower the threshold to a value where noise can be observed in control condition? It is necessary to assess the effect of Ang-II.

We have made this change (now Supplementary Fig. 6) and thank the Reviewer for the resulting improvement in clarity.

Responses to Reviewer #2

CSF movement through the brain is an important but poorly understood process. It is thought to occur through perivascular spaces (PVSs) and is potentially necessary for clearance of metabolic by-products such as amyloid- β . The authors use intravital microscopy of cranial windows in mice to follow the movement of particles injected into the cisterna magna. The authors conclude that the motion proceeds in the same direction as blood flow, and is produced by arterial pulsations within large PVSs at the brain surface. The idea that arterial pumping drives flow is not novel, but direct visualization of the process would be valuable for understanding the process. Unfortunately, the study appears very preliminary, and the PVSs shown in the figures and movies require more characterization.

We agree with the reviewer that direct visualization is valuable. We politely disagree with the adjective “preliminary”, however, given that this study involves more than 4.6 million measurements of more than 250,000 particles in 13 animals, whereas the most recent similar study tracked just 30 particles. Our measurements enabled the first calculations of Reynolds and Péclet numbers of these flows. Although the idea that arterial pumping driving flow is not novel, it had never been shown, until now. We feel that our work demonstrates a novel technology and highly quantitative approach to study PVS flow. In addition, we provide the first in-depth description of PVS fluid dynamics under controlled conditions and in acute hypertension. We agree that further characterization of PVS structure and anatomy is an excellent goal for future work; in this study, our focus is function, not structure.

Specific Comments:

In Figure 1C, D, the 3D shape of the PVS is unexpected; what does the cross section look like farther downstream, just after the bifurcation? It appears that the CFS tracers are not confined within anatomical, annular structures surrounding the arteries, but are instead trapped between the brain tissue and glass coverslip. Can the authors show that there is a biological structure containing the dextran by IHC or EM?

Indeed, the shape of the perivascular spaces we observe is somewhat unexpected. The manuscript now dedicates an entire figure to PVS shape (Fig. 2), including cross-sections both proximal and distal to the MCA bifurcation. Additionally, to ensure that coverslips were neither providing flow boundaries nor deforming brain anatomy, we performed new experiments using a thinned-skull preparation. The results are included as Supplementary Fig. 2, which shows PVS shape consistent with measurements using cranial windows. Moreover, several previous studies have shown a similar structure using transgenic β -actin and GFAP-Cre/mTmG reporter mouse lines, suggesting that the space is formed primarily by the pia mater on top and the glia limitans on bottom [Schain et al. *J. Neurosci.* 2017; Coles et al. *Methods*, 2017]. The current study focuses on the fluid dynamics of flow in the PVS, not anatomy, so we do not comment further. Because PVSs collapse during fixation, as shown in the updated version of Fig. 2 and in Supplementary Movie 1, examining PVS structure with *ex vivo* techniques like IHC or EM may not capture the *in vivo* structure.

The direction of the motion would be expected to take the particles away from the heart, into the

brain. But the particles supposedly are being drained from the brain. Can the authors explain this discrepancy?

The overarching hypothesis, supported not only by this manuscript but by prior publications [e.g. Iliff *et al. Science Translational Medicine* 2012; Ratner *et al. NeuroImage* 2017; Eide *et al. Sci. Rep.* 2018], is that fluid enters the brain through perivascular spaces around arteries, absorbs waste, and exits the brain through perivascular spaces around veins (moving away from the heart, as the reviewer notes). Our measurements are consistent with that hypothesis, and the tracer particles we inject do indicate inflow along arteries. We have added a paragraph to the manuscript that states this hypothesis explicitly and discusses its relation to our observations. No methods have yet been demonstrated for seeding perivascular spaces around veins with tracer particles that can be resolved individually (necessary for tracking), and when such methods are discovered, we will be enthusiastic to apply them.

The rationale and interpretation of Fig 1e, f are not clear. Is the yellow color dextran that has been fixed in place? Why is it only at the vessel wall, and why does it appear along the vein (when it is not visible in Fig 1c)?

We thank the reviewer for pointing out this need for clarification. The caption describing those panels (which are now Fig. 2i and Fig. 2j) now states “Overlapping lectin and dextran appear yellow.” Tracers move into the basement membranes of the vessel wall during fixation, and the manuscript now states this point explicitly, in an expanded discussion of Fig. 2. We attribute the appearance of tracer in the veins to the abnormal retrograde flow that occurs during fixation, as documented in Fig. 2 and in Supp. Video 1 and described in the manuscript.

With a bolus injection of microspheres, how long does it take to clear from the PVS? Are the pharmacokinetics reproducible?

We do not observe microspheres being cleared from the PVS, a fact which we attribute to their large size: they are sieved when fluid enters the small PVSs of penetrating arterioles and therefore cannot pass through the parenchyma nor exit along veins. Prior studies by other authors, however, have found smaller tracers are cleared from mouse brains in about 3 hours [Xie *et al. Science* 2013], from rat brains in about 3 hours [Ratner *et al. NeuroImage* 2017], and from human brains in about 24 hours [Ringstad *et al. JCI Insight* 2018].

The proposed PVS space seems to be rather large. How do the authors' measurements compare with other literature values for the width of the perivascular space?

The reviewer makes an excellent point: the PVS space is substantially larger than what can be seen *ex vivo* and has often been described in the past. However, several prior *in vivo* observations agree closely [Bedussi *et al. J Cerebr Blood F Met* 2017; Schain *et al. J. Neurosci.* 2017; Coles *et al. Methods*, 2017]. We believe that the discrepancy among prior studies was caused by the fact that PVSs collapse during fixation, and we hope that Fig. 2 and Supplementary Video 1 will help resolve confusion.

How long does it take for the particles to travel from the cisterna magna to the pial surface? Does this transport occur along arterials? Would such transport be anatomically consistent with the blood vasculature of the brain?

The manuscript now states that particles reach the MCA PVS 292 ± 26 s after the infusion begins (n=7 mice). We hypothesize that the particles do travel along arteries of the Circle of Willis between the cisterna magna and MCA, though they lie in regions of the skull that are inaccessible for *in vivo* two-photon imaging. Taking 8 mm as the approximate length of those perivascular spaces, we can roughly estimate that fluid flows through them at $(8 \text{ mm}) / (292 \text{ s}) = 27 \text{ } \mu\text{m/s}$, a speed similar to what we measure around the MCA.

Can the authors explain how peristaltic motion would be the driving force for PVS flow, when it cannot extend into the capillaries and venules? How would the particles continue moving down the vascular tree, if the flow relies on peristaltic arterial motion? Some speculation or explanation is needed here.

We thank the reviewer for pointing out this need for clarification. Blood in veins flows not because the veins are pumping, but because the heart is pumping, and the blood it pumps displaces blood that is more distal, which in turn displaces blood that is still more distal, until the blood in veins is also displaced. In other words, conservation of mass of an incompressible fluid requires uniform volume flux along the length of a closed system. We hypothesize an analogous mechanism: CSF in perivascular spaces around veins flows because it is displaced by CSF coming from perivascular spaces around arteries. The manuscript now discusses this hypothesis in substantially greater detail.

REVIEWERS' COMMENTS:

Reviewer #1 (Remarks to the Author):

I have read through the author's reply to my comments and those of reviewer #2. I consider that the authors have made adequate additions to address all criticisms that were raised. I note in particular the addition of measurements in more distal vessels and the demonstration that the unexpected PVS shape is also present in thinned skull animals.

Reviewer #2 (Remarks to the Author):

In this revised manuscript, the authors have performed additional experiments with a thinned skull preparation instead of a cranial window, and they see similar results. However, the phenomenon documented in the new experiments and the rest of this study are still not consistent with transport through the perivascular spaces surrounding vessels that has been previously characterized. My additional comments are added in-line in the rebuttal below.

Reviewer 2 original comment:

In Figure 1C, D, the 3D shape of the PVS is unexpected; what does the cross section look like farther downstream, just after the bifurcation? It appears that the CFS tracers are not confined within anatomical, annular structures surrounding the arteries, but are instead trapped between the brain tissue and glass coverslip. Can the authors show that there is a biological structure containing the dextran by IHC or EM?

Author response:

Indeed, the shape of the perivascular spaces we observe is somewhat unexpected. The manuscript now dedicates an entire figure to PVS shape (Fig. 2), including cross-sections both proximal and distal to the MCA bifurcation. Additionally, to ensure that coverslips were neither providing flow boundaries nor deforming brain anatomy, we performed new experiments using a thinned-skull preparation. The results are included as Supplementary Fig. 2, which shows PVS shape consistent with measurements using cranial windows. Moreover, several previous studies have shown a similar structure using transgenic β -actin and GFAP-Cre/mTmG reporter mouse lines, suggesting that the space is formed primarily by the pia mater on top and the glia limitans on bottom [Schain et al. J. Neurosci. 2017; Coles et al. Methods, 2017]. The current study focuses on the fluid dynamics of flow in the PVS, not anatomy, so we do not comment further. Because PVSs collapse during fixation, as shown in the updated version of Fig. 2 and in Supplementary Movie 1, examining PVS structure with ex vivo techniques like IHC or EM may not capture the in vivo structure.

Reviewer 2 follow-up comment:

The issue seems to be an inconsistent definition of PVS. The authors write "our in vivo measurements reveal large PVSs surrounding pial arteries." But here, they write "the space is formed primarily by the pia mater on top and the glia limitans on bottom." The authors need to be clear about whether the confining structure is a "sleeve" surrounding each vessel, or a space confined from top and bottom, parallel to the skull. The images and videos would support the latter, but not the former. For example, there are many vessels within the plane of focus of their preparations, but the only "PVS" visible is that of the large Y-shaped artery and its branches. It seems there is no space left for the PVSs of all the other vessels in the image, and rather, the fluid is moving in a large, planar space that engulfs all vessels. If this is the case, the destination of the flow – and thus its clearance – is not obvious.

This discrepancy is most obvious in Supplementary Movie 3. This movie shows particles moving in trajectories inconsistent with PVS confinement. It is difficult to imagine how the particles are moving within PVS in the region after the bifurcation—the particles approach from above to cross over the upper daughter branch, and then continue toward the bottom of the screen, in trajectories that should presumably be crossing from the PVS of the upper branch to that of the lower branch. The authors should create an image of all the composited trajectories and then sketch on top the putative location of the PVS confining structure.

Reviewer 2 original comment:

The rationale and interpretation of Fig 1e, f are not clear. Is the yellow color dextran that has been fixed in place? Why is it only at the vessel wall, and why does it appear along the vein (when it is not visible in Fig 1c)?

Author response:

We thank the reviewer for pointing out this need for clarification. The caption describing those panels (which are now Fig. 2i and Fig. 2j) now states “Overlapping lectin and dextran appear yellow.” Tracers move into the basement membranes of the vessel wall during fixation, and the manuscript now states this point explicitly, in an expanded discussion of Fig. 2. We attribute the appearance of tracer in the veins to the abnormal retrograde flow that occurs during fixation, as documented in Fig. 2 and in Supp. Video 1 and described in the manuscript.

Reviewer 2 follow-up comment:

It is not clear what the authors mean by “retrograde flow,” as reversal of flow within a peri-arterial PVS would not move the lectin towards veins. Instead, this result is consistent with the lectin attaching to all basement membrane structures within an extravascular 2D space that contains both the artery and vein.

Reviewer 2 original comment:

With a bolus injection of microspheres, how long does it take to clear from the PVS? Are the pharmacokinetics reproducible?

Author response:

We do not observe microspheres being cleared from the PVS, a fact which we attribute to their large size: they are sieved when fluid enters the small PVSs of penetrating arterioles and therefore cannot pass through the parenchyma nor exit along veins. Prior studies by other authors, however, have found smaller tracers are cleared from mouse brains in about 3 hours [Xie et al. *Science* 2013], from rat brains in about 3 hours [Ratner et al. *NeuroImage* 2017], and from human brains in about 24 hours [Ringstad et al. *JCI Insight* 2018].

Reviewer 2 follow-up comment:

This is interesting. Can the authors quantify this sieving to show the particles get trapped and accumulate around descending arterioles? This would support the assertion that the authors are observing PVS flow.

Reviewer 2 original comment:

The proposed PVS space seems to be rather large. How do the authors' measurements compare with other literature values for the width of the perivascular space?

Author response:

The reviewer makes an excellent point: the PVS space is substantially larger than what can be seen *ex vivo* and has often been described in the past. However, several prior *in vivo* observations agree closely [Bedussi et al. *J Cerebr Blood F Met* 2017; Schain et al. *J. Neurosci.* 2017; Coles et al. *Methods*, 2017]. We believe that the discrepancy among prior studies was caused by the fact that PVSs collapse during fixation, and we hope that Fig. 2 and Supplementary Video 1 will help resolve confusion.

Reviewer 2 follow-up comment:

Have the authors tried to label the PVS structure directly in the live animal by injecting primary labeled antibodies to GFAP or AQP4 ? If the microspheres can access the PVS, then antibodies (which are much smaller) should also, and the anatomical structure could be visualized via intravital microscopy, without fixation.

Reviewer 2 original comment:

Can the authors explain how peristaltic motion would be the driving force for PVS flow, when it cannot extend into the capillaries and venules? How would the particles continue moving down the vascular tree, if the flow relies on peristaltic arterial motion? Some speculation or explanation is needed here.

Author response:

We thank the reviewer for pointing out this need for clarification. Blood in veins flows not because the veins are pumping, but because the heart is pumping, and the blood it pumps displaces blood that is more distal, which in turn displaces blood that is still more distal, until the blood in veins is also displaced. In other words, conservation of mass of an incompressible fluid requires uniform volume flux along the length of a closed system. We hypothesize an analogous mechanism: CSF in perivascular spaces around veins flows because it is displaced by CSF coming from perivascular spaces around arteries. The manuscript now discusses this hypothesis in substantially greater detail.

Reviewer 2 follow-up comment:

If the driving force is pressure produced from arterial motion, then how is it possible for some particles to travel in the opposite direction of arterial flow, such as after the bifurcation in Supplementary Movie 2?

Reviewer #3 (Remarks to the Author):

Comments to Authors:

Concerning the reservations of Rev. 2, I concur with the authors that this is a CSF/ISF flow paper and not an anatomical description of the perivascular and subarachnoid spaces, *i.e.*, size, shape, etc., which the techniques used are not best equipped to address. The peristaltic flow is convincingly shown in the movies and the impact of perfusion-fixation is an important contribution, since such shrinkage was hypothesized but never shown in such a dramatic manner as in movie 1.

However, I concur with Rev. 2 that the weak link of the paper concerns the fact that the particles do not seem to enter the brain parenchyma along penetrating vessels. Therefore, it remains unclear whether the flow reaches arterioles and capillaries within the brain, where the clearance of the accumulated waste produced by the brain would take place, as hypothesized. The crossing of the flow from the arterial to the venous side is a related problem. I would recommend that the authors

recognize the knowledge gap and tone down some of the statement about a role of this system in amyloid clearance etc. This is also appropriate because of the resistance that the field is showing in embracing the “glymphatic” model (e.g., *Acta Neuropathologica* (2018) 135: 38; *eLife* 2017;6:e27679) that cannot be ignored.

Another aspect of the study that needs a more cautious presentation concerns the effect of hypertension on particle flow. While the observation of reduced CSF flow is consistent with imaging and pathological data suggesting that chronic hypertension promotes amyloid accumulation, acute elevations of blood pressure produced by pharmacological doses of angiotensin II, popular up to the 1980s, are hardly representative of human essential hypertension. There are more chronic models (“slow pressor” angiotensin hypertension, BPH mice; see *JCI* 126:4674, 2016) that are currently thought to be more translationally relevant because the blood pressure elevation is gradual and is associated with structural changes in brain vessels (remodeling, hypertrophy, stiffening, etc.), thought to play a major role in the altered brain proteostasis observed in human hypertension.

Responses to Reviewer #1

I have read through the author's reply to my comments and those of reviewer #2. I consider that the authors have made adequate additions to address all criticisms that were raised. I note in particular the addition of measurements in more distal vessels and the demonstration that the unexpected PVS shape is also present in thinned skull animals.

We thank Reviewer #1 for these positive comments.

Responses to Reviewer #2

Reviewer 2 original comment:

In Figure 1C, D, the 3D shape of the PVS is unexpected; what does the cross section look like farther downstream, just after the bifurcation? It appears that the CFS tracers are not confined within anatomical, annular structures surrounding the arteries, but are instead trapped between the brain tissue and glass coverslip. Can the authors show that there is a biological structure containing the dextran by IHC or EM?

Author response:

Indeed, the shape of the perivascular spaces we observe is somewhat unexpected. The manuscript now dedicates an entire figure to PVS shape (Fig. 2), including cross-sections both proximal and distal to the MCA bifurcation. Additionally, to ensure that coverslips were neither providing flow boundaries nor deforming brain anatomy, we performed new experiments using a thinned-skull preparation. The results are included as Supplementary Fig. 2, which shows PVS shape consistent with measurements using cranial windows. Moreover, several previous studies have shown a similar structure using transgenic β -actin and GFAP-Cre/mTmG reporter mouse lines, suggesting that the space is formed primarily by the pia mater on top and the glia limitans on bottom [Schain et al. J. Neurosci. 2017; Coles et al. Methods, 2017]. The current study focuses on the fluid dynamics of flow in the PVS, not anatomy, so we do not comment further. Because PVSs collapse during fixation, as shown in the updated version of Fig. 2 and in Supplementary Movie 1, examining PVS structure with ex vivo techniques like IHC or EM may not capture the in vivo structure.

Reviewer 2 follow-up comment:

The issue seems to be an inconsistent definition of PVS. The authors write “our in vivo measurements reveal large PVSs surrounding pial arteries.” But here, they write “the space is formed primarily by the pia mater on top and the glia limitans on bottom.” The authors need to be clear about whether the confining structure is a “sleeve” surrounding each vessel, or a space confined from top and bottom, parallel to the skull. The images and videos would support the latter, but not the former. For example, there are many vessels within the plane of focus of their preparations, but the only “PVS” visible is that of the large Y-shaped artery and its branches. It seems there is no space left for the PVSs of all the other vessels in the image, and rather, the fluid is moving in a large, planar space that engulfs all vessels. If this is the case, the destination of the flow – and thus its clearance – is not obvious.

This discrepancy is most obvious in Supplementary Movie 3. This movie shows particles moving in trajectories inconsistent with PVS confinement. It is difficult to imagine how the particles are moving within PVS in the region after the bifurcation—the particles approach from above to cross over the upper daughter branch, and then continue toward the bottom of the screen, in trajectories that should presumably be crossing from the PVS of the upper branch to that of the lower branch. The authors should create an image of all the composited trajectories and then sketch on top the putative location of the PVS confining structure.

This manuscript focuses on fluid flow, and we welcome future work that can provide a careful characterization of the specific structures that bound that fluid flow, which will likely require different methods. We have included that sentiment in the manuscript: “Identifying the specific structures that bound the observed PVSs is a worthy topic for future study.”

However, the reviewer's description of the PVS as "a large, planar space that engulfs all vessels" is inconsistent with our observations using both dye and tracer particles, as shown in Figures 1 and 2, Supplementary Figures 2, 5, and 7, and Supplementary Movies 1, 2, and 3. We observe that just distal to bifurcations, PVSs are wider, and flow there is disordered, but at slightly more distal locations, separate arteries have separate PVSs (see especially Fig. 2c).

Reviewer 2 original comment:

The rationale and interpretation of Fig 1e, f are not clear. Is the yellow color dextran that has been fixed in place? Why is it only at the vessel wall, and why does it appear along the vein (when it is not visible in Fig 1c)?

Author response:

We thank the reviewer for pointing out this need for clarification. The caption describing those panels (which are now Fig. 2i and Fig. 2j) now states "Overlapping lectin and dextran appear yellow." Tracers move into the basement membranes of the vessel wall during fixation, and the manuscript now states this point explicitly, in an expanded discussion of Fig. 2. We attribute the appearance of tracer in the veins to the abnormal retrograde flow that occurs during fixation, as documented in Fig. 2 and in Supp. Video 1 and described in the manuscript.

Reviewer 2 follow-up comment:

It is not clear what the authors mean by "retrograde flow," as reversal of flow within a peri-arterial PVS would not move the lectin towards veins. Instead, this result is consistent with the lectin attaching to all basement membrane structures within an extravascular 2D space that contains both the artery and vein.

As discussed above, our observations are inconsistent with flow through spaces that contain both the artery and vein.

Reviewer 2 original comment:

With a bolus injection of microspheres, how long does it take to clear from the PVS? Are the pharmacokinetics reproducible?

Author response:

We do not observe microspheres being cleared from the PVS, a fact which we attribute to their large size: they are sieved when fluid enters the small PVSs of penetrating arterioles and therefore cannot pass through the parenchyma nor exit along veins. Prior studies by other authors, however, have found smaller tracers are cleared from mouse brains in about 3 hours [Xie et al. Science 2013], from rat brains in about 3 hours [Ratner et al. NeuroImage 2017], and from human brains in about 24 hours [Ringstad et al. JCI Insight 2018].

Reviewer 2 follow-up comment:

This is interesting. Can the authors quantify this sieving to show the particles get trapped and accumulate around descending arterioles? This would support the assertion that the authors are observing PVS flow.

We agree with the reviewer that the sieving of tracers is an interesting phenomenon that gives insight into the flow path of cerebrospinal fluid. The manuscript now states that idea: "The exact path by which CSF flows into the deeper brain is an important topic of future research; its study

will require new methods because particles large enough to be tracked individually are not transported along the PVSs of penetrating arterioles.”

Reviewer 2 original comment:

The proposed PVS space seems to be rather large. How do the authors' measurements compare with other literature values for the width of the perivascular space?

Author response:

The reviewer makes an excellent point: the PVS space is substantially larger than what can be seen ex vivo and has often been described in the past. However, several prior in vivo observations agree closely [Bedussi et al. J Cerebr Blood F Met 2017; Schain et al. J. Neurosci. 2017; Coles et al. Methods, 2017]. We believe that the discrepancy among prior studies was caused by the fact that PVSs collapse during fixation, and we hope that Fig. 2 and Supplementary Video 1 will help resolve confusion.

Reviewer 2 follow-up comment:

Have the authors tried to label the PVS structure directly in the live animal by injecting primary labeled antibodies to GFAP or AQP4? If the microspheres can access the PVS, then antibodies (which are much smaller) should also, and the anatomical structure could be visualized via intravital microscopy, without fixation.

This is a very interesting idea, we have never attempted *in vivo* antibody labeling of the PVS structures. We would speculate that labeling might be limited by the amount of antibody that can reach an individual PVS and whether it is enough signal to detect via multiphoton microscopy. An alternative approach is to use cell-specific fluorescent reporter lines (e.g. GFAP-Cre/mTmG) to identify the location of the astrocyte endfoot, vascular smooth muscle cell, etc. which we plan to do in future studies [Schain et al. J. Neurosci. 2017; Coles et al. Methods, 2017].

Reviewer 2 original comment:

Can the authors explain how peristaltic motion would be the driving force for PVS flow, when it cannot extend into the capillaries and venules? How would the particles continue moving down the vascular tree, if the flow relies on peristaltic arterial motion? Some speculation or explanation is needed here.

Author response:

We thank the reviewer for pointing out this need for clarification. Blood in veins flows not because the veins are pumping, but because the heart is pumping, and the blood it pumps displaces blood that is more distal, which in turn displaces blood that is still more distal, until the blood in veins is also displaced. In other words, conservation of mass of an incompressible fluid requires uniform volume flux along the length of a closed system. We hypothesize an analogous mechanism: CSF in perivascular spaces around veins flows because it is displaced by CSF coming from perivascular spaces around arteries. The manuscript now discusses this hypothesis in substantially greater detail.

Reviewer 2 follow-up comment:

If the driving force is pressure produced from arterial motion, then how is it possible for some particles to travel in the opposite direction of arterial flow, such as after the bifurcation in Supplementary Movie 2?

The reviewer raises a great question. The disordered flow in bifurcations is an interesting topic for future research, not least because plaques often accumulate there. Peristaltic pumping is known to cause recirculation, especially in regions with large cross-sections; Jaffrin and Shapiro provide an excellent overview (Jaffrin, M. Y. & Shapiro, A. H. Peristaltic pumping. *Annu. Rev. Fluid Mech.* **3**, 13–37, 1971). We have expanded the manuscript’s discussion of flow in bifurcations: “The likely reason flows inside arterial bifurcations are often stagnant is that perivascular pumping generates opposing pressure gradients that sum to approximately zero in these regions. Small differences in perivascular pumping strength between each daughter vessel may drive slow reverse flow (toward more proximal locations) locally in this region. We have observed substantial reverse flow in only one bifurcation region of one experiment; in this case, the daughter branches had significantly different diameters, suggesting the difference in perivascular pumping strength may have been considerable. Flow in the bifurcation, typically slow and complicated, deserves further study; still, flow in PVSs overwhelmingly proceeds toward more distal locations.”

Responses to Reviewer #3

Concerning the reservations of Rev. 2, I concur with the authors that this is a CSF/ISF flow paper and not an anatomical description of the perivascular and subarachnoid spaces, i.e., size, shape, etc., which the techniques used are not best equipped to address. The peristaltic flow is convincingly shown in the movies and the impact of perfusion-fixation is an important contribution, since such shrinkage was hypothesized but never shown in such a dramatic manner as in movie 1.

We thank the reviewer for these words of support.

*However, I concur with Rev. 2 that the weak link of the paper concerns the fact that the particles do not seem to enter the brain parenchyma along penetrating vessels. Therefore, it remains unclear whether the flow reaches arterioles and capillaries within the brain, where the clearance of the accumulated waste produced by the brain would take place, as hypothesized. The crossing of the flow from the arterial to the venous side is a related problem. I would recommend that the authors recognize the knowledge gap and tone down some of the statement about a role of this system in amyloid clearance etc. This is also appropriate because of the resistance that the field is showing in embracing the “glymphatic” model (e.g., *Acta Neuropathologica* (2018) 135:38; *eLife* 2017;6:e27679) that cannot be ignored.*

We agree with the reviewer that the path of CSF flow beyond PVSs of surface arteries is a key topic for future study, and the manuscript now states that idea: “The exact path by which CSF flows into the deeper brain is an important topic of future research; its study will require new methods because particles large enough to be tracked individually are not transported along the PVSs of penetrating arterioles.”

*Another aspect of the study that needs a more cautious presentation concerns the effect of hypertension on particle flow. While the observation of reduced CSF flow is consistent with imaging and pathological data suggesting that chronic hypertension promotes amyloid accumulation, acute elevations of blood pressure produced by pharmacological doses of angiotensin II, popular up to the 1980s, are hardly representative of human essential hypertension. There are more chronic models (“slow pressor” angiotensin hypertension, BPH mice; see *JCI* 126:4674, 2016) that are currently thought to be more translationally relevant because the blood pressure elevation is gradual and is associated with structural changes in brain vessels (remodeling, hypertrophy, stiffening, etc.), thought to play a major role in the altered brain proteostasis observed in human hypertension.*

We agree with the reviewer that angiotensin II is not a model for chronic hypertension. Acute hypertension was used in this study to gain insight into acute changes of the pumping mechanism. As the reviewer points out, chronic hypertension is associated with remodeling of the vasculature and also of surrounding tissue. We have further clarified that point in the manuscript: “Future studies, using methods that better model human essential hypertension (spontaneous hypertensive mice or slow pressor angiotensin II infusion) will provide insight into how vascular remodeling in response to long-lasting elevation of blood pressure affect PVS fluid transport.”